# Potential Epigenetic Impacts of Phytochemicals on Ruminant Health and Production: Connecting Lines of Evidence

**DOI:** 10.3390/ani15121787

**Published:** 2025-06-17

**Authors:** Sebastian P. Schreiber, Juan Villalba, Mirella L. Meyer-Ficca

**Affiliations:** 1Department of Wildland Resources, Utah State University, Logan, UT 84322, USA; s.schreiber-pan@usu.edu; 2Department of Veterinary, Clinical and Life Sciences, Utah State University, Logan, UT 84322, USA; mirella.meyer@usu.edu

**Keywords:** antioxidants, secondary metabolites, fetal programming, gene expression, livestock, nutrition, genetics, heritable

## Abstract

Phytochemicals are secondary compounds produced by plants that cannot be classified as macronutrients, vitamins, or minerals. These highly diverse and numerous compounds are non-essential for plant function or for animal nutrition, but they are now associated with many health benefits in animals and humans. Grazing animals often select for small doses of these compounds to improve their health in a process known as self-medication. Many phytochemicals are powerful antioxidants due to their structure. Other benefits of phytochemicals include anti-inflammatory, anti-cancer, and immune-boosting effects. These are likely related to epigenetic processes, i.e., changes in gene expression. If this is so, as many human studies suggest, major implications for ruminant livestock production exist. This is because the epigenetic marks that determine an animal’s traits are potentially heritable. An example of this phenomenon is termed “fetal programming”. Ultimately, environmental conditions, in this case phytochemicals within diets, can produce long-lasting effects in individuals and even their offspring, a revolutionary concept for an industry that relies on the breeding of animals. While this is still an emerging topic, we explore the existing research and suggest avenues for future research to discover the extent to which phytochemicals can be used to improve animal health and production.

## 1. Introduction

The emerging field of epigenetics has many implications for ruminant health and productivity. Epigenetics investigates the regulation of gene expression through reversible modifications to chromatin structure and DNA, independent of and generally without any changes to the underlying genetic sequence. These epigenetic modifications, including DNA methylation, histone modifications (including acetylation and methylation), and chromatin remodeling, can activate or silence genes and profoundly influence cellular function [1,2].

The term “epigenetics” describes how systems of chromatin marks and structural changes, mediated by chemical modifications of DNA, histone proteins, and the resulting chromatin structure, determine which genes encoded by the genome are expressed or repressed. It comprises mechanisms like DNA methylation, histone modifications (such as acetylation, methylation, phosphorylation, and ubiquitination), and higher-order chromatin remodeling. The phenotype of any individual is influenced significantly by their epigenome, which in turn can be influenced by the individual’s environment [3]. Such environmental impacts can also be conferred upon offspring and may potentially persist for multiple generations. Thus, epigenetic mechanisms represent potential tools to ensure health and productivity in individual animals as well as their offspring. Persistent epigenome modifications have been consistently observed to respond to dietary interventions in humans [4] and various animal models, including rodents [5] and livestock [6]. The epigenetic effects of diet have largely been examined under the framework of macronutrient and calorie excesses or deficiencies. However, researchers are now beginning to recognize the health benefits of consuming diets rich in diverse phytochemicals for both ruminants and humans [7,8,9]. While there are some emerging studies exploring the impact of isolated phytochemicals on the epigenome [10,11], the potential impacts of overall dietary phytochemical diversity on the epigenome of individuals and their offspring have been largely unexplored, especially in ruminant livestock. The current paper aims to review the literature surrounding this topic.

## 2. Methods

Because direct evidence for the epigenetic effects of phytochemicals on ruminant health and production is currently limited, we structured this article as a narrative review that outlines the foundational concepts of epigenetic mechanisms, health benefits of phytochemicals, and how they might interact to influence traits across generations. We selected both recent and seminal studies in the literature relevant to our narrative, which aims to provide researchers within the fields of plant secondary chemistry, ruminant nutrition, and molecular (epi)genetics with the necessary background to understand the mechanisms and potential implications of the emerging interactions between these fields. Thus, these sections are not based on an exhaustive or otherwise systematic literature search.

We conducted a more in-depth literature search to identify studies relevant to actual epigenetic effects of phytochemicals within ruminants, which are summarized within Section 6 (Existing Evidence: Epigenetic Effects of Phytochemicals), primarily Section 6.4 (Epigenetic Effects of Phytochemicals in Ruminants) and Section 6.5 (Transgenerational Effects of Phytochemicals in Ruminants). Our goal was to synthesize evidence from randomized controlled trials (RCTs) or other relevant experiments that directly assessed how dietary exposure to plant secondary compounds influences epigenetic mechanisms and phenotypic outcomes published in English-language articles available up to May 2025. We searched only for peer-reviewed articles published within indexed academic journals in the following databases: Google Scholar, JSTOR, EBSCOhost, PubMed, and ScienceDirect. We primarily used the following keyword phrases: “epigenetic effects of phytochemicals in ruminants”, “epigenetic effects of secondary compounds in ruminants”, “effects of phytochemicals on gene expression in ruminants”, “effects of secondary compounds on gene expression in ruminants”, “phytochemicals ruminant fetal programming”, and “secondary compounds ruminant fetal programming”. Reference lists of key studies were also scanned for additional relevant publications. In addition, we monitored new publications using Google Scholar’s article alert notifications for the keywords “livestock epigenetics” beginning approximately in summer 2023. Studies were excluded if they lacked both phenotypic and epigenetic outcome data, if the phytochemical intervention was not adequately characterized, or if the study species was not a ruminant or otherwise relevant to ruminant biology. The resulting studies are synthesized and discussed narratively in Section 6.

## 3. Phytochemicals and Animal Health

Plant secondary compounds (PSCs), such as phenolics, terpenoids, and alkaloids, were initially regarded as waste products of plant metabolism due to their non-essential nature when they were first identified in the second half of the 19th century [12,13,14]. Since then, many different roles have been recognized for these chemicals, especially as defense compounds against herbivory [14,15]. Thus, within ruminant nutrition, PSCs have long been regarded as primarily anti-nutritional compounds [16]. Indeed, PSCs often reduce the amount of plant tissue ingested by herbivores by inducing aversive post-ingestive feedback [17]. PSCs can act on multiple cellular and metabolic processes, cause reductions in digestibility, and may result in weight loss and even death [18]. However, PSCs protect plants from many threats other than herbivory, such as the attenuation of UV-induced oxidative stress, reductions in parasitic burdens and microbial infections, and the amelioration of harmful abiotic conditions [19,20,21,22]. These beneficial effects can also manifest in consumers within higher trophic levels, such as herbivores consuming plant material containing these PSCs in appropriate doses. For instance, the consumption of various PSCs can produce antiparasitic [23,24,25], antimicrobial [26,27], anti-inflammatory [28,29,30], antioxidative [31,32], and immunomodulatory effects [33,34,35]. This phenomenon is predicted by co-evolution theory, stating that herbivores that can not only tolerate but derive benefit from low doses of PSCs would have enhanced fitness [36]. Because the beneficial or harmful effects of PSCs are dose-dependent [37,38], the concept of “hormesis” is also relevant for understanding this dynamic [39]. In the context of nutrition and potential health benefits, PSCs are often referred to as phytochemicals, which we will use in the remainder of this review. We will also occasionally use the term “secondary metabolites” to refer collectively to phytochemicals and their downstream metabolites within the body of consumers.

Ultimately, the properties and potential benefits of consuming any phytochemical-containing plant depend on the specific chemical composition, structure, and concentrations of the phytochemicals in plant tissues, as well as their interactions with other plant biochemicals and animal tissues [40]. Thus, selection for beneficial doses of phytochemicals may be a primary driver of diet diversity in herbivores [7,41,42]. Herbivores typically select 3 to 5 plants for the bulk of any one meal but often consume small amounts of 50 to 75 plants over the course of a day when foraging in diverse plant communities [43]. Taxonomic diversity contributes to a rich variety of phytochemicals, with distinct types and profiles of phytochemicals identified across herbs, shrubs, and grasses [44]. To date, over 200,000 unique phytochemical structures have been identified [12]. The same post-ingestive feedback mechanisms that regulate primary nutrient intake are responsible for the observed preferences for diverse phytochemical-containing feeds and forages [17,45]. This may explain observations of animals consuming chemically defended or otherwise unpalatable plants even when higher-quality forages are available for consumption [35]. Arguably, the same applies to the preferences of humans for foods low in primary nutrients but rich in phytochemicals, such as teas, coffee, spices, and herbs [43]. This phenomenon is especially pronounced in sick animals, where ingestion of these medicinal compounds in appropriate doses as prophylactic or therapeutical self-medication can move their physiology towards a healthy homeostatic state [42,46,47]. Phytochemicals, especially polyphenols, are widely recognized for their role as powerful antioxidants due to their chemical structure [31,48,49]. However, polyphenols and other phytochemicals have been linked with a myriad of additional health benefits in both humans [50] and ruminant livestock [42], the mechanisms of which are generally less well understood. Emerging evidence suggests that these beneficial effects, including anti-inflammatory, immunomodulatory, and anticarcinogenic effects, are related to phytochemicals’ role in signaling and modulating gene expression [51,52,53].

## 4. Epigenetics: Foundational Concepts

### 4.1. Epigenetic Mechanisms

Epigenetics can be defined as a “stably heritable phenotype resulting from changes in a chromosome without alterations in the DNA sequence” and “the structural adaptation of chromosomal regions so as to register, signal or perpetuate altered activity states” [54,55]. The epigenome, a collective term for all epigenetic information, determines the likelihood that any one gene or region of the genome will be expressed [54]. The epigenome is also tissue-specific to ensure that only genes required for a respective tissue are expressed in those cells [54]. Recent studies in humans and other animals have implicated epigenetic mechanisms in nearly all aspects of their biology [56,57,58,59], with epigenetic processes playing fundamental roles in cell differentiation and tissue development, metabolism, circadian cycles, health and immunity, memory and cognition, and aging and lifespan, as well as the pathogenesis of most diseases and cancers [60].

Epigenetic information is stored and transmitted in the form of hierarchical and often interrelated layers of chromatin modifications and structural changes [61]: The finest scale of these layers is the methylation of cytosine bases within DNA strands, with increased methylation generally resulting in the repression of genes [62]. The next layer entails the spacing of nucleosomes, with more sparsely spaced nucleosomes equating to DNA that is more accessible and more likely to be transcribed. The third layer consists of the post-translational histone modifications (PTMs) of proteins, mostly the methylation, acetylation, phosphorylation, and ubiquitination (as well as other less-understood PTMs) of amino acid residues within histones, primarily within the tail region of histones H3 and H4 [60,62,63]. The affinity for transcription factors to bind to DNA, especially at the transcription start site of genes, is also considered a layer of epigenetic information [60,62,63]. The largest-scale layer is the three-dimensional structure of chromatin [60,62,63]. Chromatin, which is associated with or proximal to the nuclear lamina (lamina-associated domains, LADs), is mostly inactive, while chromatin within loop-like shapes located closer towards the center of a nucleus (known as topologically associated domains, TADs) is more transcriptionally active [61]. Activity within one of these layers often affects activity within others, resulting in varying levels of redundant, or, at times, conflicting signals that ultimately collaborate to precisely fine-tune gene expression [64]. The interaction between these layers of epigenetic control mechanisms can produce highly and indefinity repressed “constitutive heterochromatin”, actively expressed “euchromatin”, or a more flexible “facultative heterochromatin”, in which genes can be “poised” for future expression [61]. Notably, non-coding RNA molecules (ncRNAs) also serve as additional regulators of gene expression on many of the aforementioned levels, e.g., by binding or interacting with DNA and chromatin modifiers [65].

Of these many layers of epigenetic information, cytosine methylation is the most well understood. It is also the most stable over time and can persist through DNA replication, because cytosine–guanine sequences (CpGs) are “mirrored” on the complementary strands of DNA, which enables DNA methyltransferase DNMT1 enzymes to recognize which CpGs are hemi-methylated following replication and copy the methylation pattern onto the new DNA strand. Thus, CpG methylation is a form of epigenetic information that is stably inherited through cell division [66]. For this reason, CpG methylation is a focus of research into potentially heritable individual-specific epigenomic patterns [67].

### 4.2. Environmental Influences and Their Heritability

While estimates vary, DNA in mammalian sperm cells is generally highly methylated at approximately 90%, whereas oocytes more closely resemble somatic cells at about 60% methylation [11,68]. When these combine to form a zygote, this proportion declines precipitously during the pre-implantation phase to as low as 30% [11,69]. As the early embryo matures and implants into the uterine lining, cells begin to differentiate into separate lineages and acquire the cell-type-specific epigenetic patterns, resulting in a dramatic increase in the total percentage of methylated CpGs as well as the amount of histone marks [68,69]. During these early stages of development, the epigenome of the developing organism is particularly sensitive to environmental conditions [11,69]. Thus, in utero environmental exposures can influence an offspring’s epigenetic marks, including methylation patterns, which ultimately affect that individual’s phenotype. This process is known as “fetal programming” and forms the basis for the concept of Developmental Origins of Health and Disease (DOHaD).

Interestingly, this phenomenon not only affects the developing organism but also their future germ cells, as the primordial germ cell (PGC) lineage begins to develop during these early stages as well. After the hypomethylated period of the early embryo, a subset of inner cell mass cells begins to differentiate into PGCs. PGCs in bovines have been identified in the embryo as early as day 16 post-fertilization [70]. Once differentiated, PGCs undergo a second wave of widespread demethylation and re-methylation, reaching as low as 5% methylation. Bovine PGCs begin to migrate by day 20, at which time the initiation of epigenetic remodeling is already evident [70]. Thus, periconceptional environmental stimuli have the potential to impact cells of all three generations that are present in a pregnant individual. The small minority (approximately 30%) [11,69] of methylated CpG sites that persist through the pre-implantation hypomethylated state is thought to serve as the basis for the direct inheritance of epigenetic information from the parental generation (P) to the first filial generation (F1) [71]. Epigenetic marks, which are eventually inherited from the P to the F2 generation (via the germ cells of F1), must also persist through the second wave of demethylation that occurs during germ cell development. While the periconceptional period is disproportionately important for epigenome establishment, epigenetic changes can occur at any point in an individual’s life in both somatic and germ cells [58].

In summary, information about the environment can be transmitted into an individual’s epigenome in a variety of ways [55,72] (Figure 1). Listed in order from the transmission of the most recent environmental information to the transmission of the most historically distant information, these mechanisms include the following: (1) An epigenetic change might arise in an individual in response to a direct environmental condition or downstream endogenous signal. (2) This direct exposure can occur in utero – fetal programming. Early embryonic development is a hypersensitive state during which many long-term methylation patterns are established. (3) An individual can inherit parental epigenetic marks (if those marks persisted through the pre-implantation demethylation phase) that parental germ cells acquired either from their direct environment or during their own fetal programming phase. (4) An individual can inherit grand-parental epigenetic marks (if those marks persist through the pre-implantation demethylation phase, the PGC demethylation phase, and any demethylation that may occur later in life) that parental germ cells inherited from their parents during embryogenesis [73]. Additionally, epigenetic changes can arise purely stochastically, for reasons unrelated to the external environment or endogenous signals. Such aberrant marks accrue with age [59]. Emerging research has identified changes in the methylation of specific genes as a primary hallmark of aging and maturation in cattle [74]. Thus, epigenetic changes are not inherently positive or negative; they can transmit both adaptive and maladaptive information and can likewise lead to the optimal regulation of biological processes or cause widespread dysregulation.

### 4.3. The Viable Yellow Agouti Mouse Model

The viable yellow agouti mice model is a common and early framework for examining the phenomenon of fetal programming, especially the effects of parental diet on offspring phenotype [75]. Agouti viable yellow mice possess a viral retrotransposon, IAP (intracisternal A particle), that is located just upstream of the mammalian melanin controlling gene, *Asip* (agouti signaling protein). IAP promotes widespread expression of the *Asip* gene, rather than expression only in hair follicle cells. Interestingly, this ectopic expression of the *Asip* gene not only produces the eponymous yellow coat but also impedes satiety signaling, which leads to obesity, diabetes, and an elevated risk of cancer [76]. The IAP promoter is also subject to varying degrees of CpG methylation. This degree of methylation determines the A^vy^/a individual’s phenotype, with hypomethylation producing the yellow, obese “agouti” phenotype, hypermethylation producing the dark brown and lean “pseudo-agouti” (wild type) phenotype, and partial methylation often producing mottled or intermediate individuals [75,76]. Thus, A^vy^/a mothers supplemented with methyl-donor-rich foods produce offspring of the pseudo-agouti phenotype in greater proportions than A^vy^/a mothers who were not supplemented [75].

The obvious phenotypic readout made the agouti viable yellow mouse an invaluable and relevant research model to elucidate the basic epigenetic mechanisms, CpG methylation in this case, that connect maternal nutrition to offspring phenotype and ultimately health. While this seminal case study is specific to mice possessing the IAP retrotransposon, the principle that de novo CpG methylation during early development is especially sensitive to environmental conditions, specifically maternal diet, and leads to relatively stable phenotypic changes in the offspring is applicable to all mammals, including ruminants [11,58,68,69]. Furthermore, because supplementation of the phytochemical genistein also induces IAP methylation, this model was amongst the first to demonstrate that phytochemical consumption can cause epigenetic changes, in this case, likely via the regulation of certain chromatin modifiers [76].

## 5. Existing Evidence: Epigenetic Effects of Nutritional Status

### 5.1. Epigenetic Effects of Nutrition in Individuals

Because the epigenome and metabolome are highly interconnected [77,78,79], diet is a very important factor regulating an animal’s epigenome. An organism’s metabolome comprises the complete set of small-molecule chemicals present in that organism’s tissues, and it is the product of both their nutrition (i.e., foods consumed), including both essential nutrients as well as non-essential compounds, including phytochemicals, and their metabolism (i.e., the activity of metabolic pathways by which those foods are processed), which converts ingested nutrients and compounds into a plethora of downstream metabolites. Many of these metabolites can either directly or indirectly affect the epigenome and thereby modulate gene expression [78]. Several metabolites that are related to the metabolism of macro- and micronutrients, ubiquitous and essential throughout the body, are also common cofactors for various epigenetic functions. Examples include adenosine triphosphate (ATP), nicotinamide adenine dinucleotide (NAD^+^), Acetyl-CoA, S-Adenosylmethionine (SAM), and α-ketoglutarate (αKG).

SAM, a metabolite synthesized in the body from methionine (an essential amino acid obtained from dietary sources) and ATP by the “one-carbon metabolism” pathway, serves as an essential cofactor for DNA and RNA methylation, where it serves as the primary methyl-group donor for over 200 methyltransferase enzymes [80,81]. Thus, within the viable yellow agouti mouse model, individuals’ ability to methylate the IAP promoter is directly related to their respective intake of vitamins B9 and B12, which are essential for the one-carbon pathway that produces SAM [82]. A similar phenomenon has also been replicated in cattle supplemented with the methyl-group donor methionine [83,84]. Gene expressions of dairy cattle were markedly altered by methionine supplementation, especially in genes associated with the innate and adaptive immune system [83]. In beef cattle, the same mechanisms produced phenotypic benefits associated with milk production, digestibility, and feed efficiency [84]. Additionally, metabolites that modulate cell energy balance play major epigenetic roles. For example, during high-energy states, the acetyl-donor molecule Acetyl-Coenzyme A is abundant relative to Coenzyme A (a derivative of vitamin B5). This results in increased histone acetyltransferase (HAT) activity and increased chromatin accessibility, leading to lipogenesis and adipocyte differentiation [78]. Conversely, during low-energy states, adenosine monophosphate (AMP) is abundant relative to ATP, and NAD^+^ is abundant relative to NADH (derivative of B3 vitamins). These shifts cause an increase in histone phosphorylation and an increase in sirtuin (a family of histone deacetylase enzymes) activity, respectively [78]. Thus, chromatin modifiers sense information about the body’s nutrient and energy status, transmitted by the metabolome, and utilize that information to program the epigenome and thereby modify both short- and long-term patterns of gene expression [60,78]. While this conclusion is supported by lines of evidence from humans, rodents, ruminants, and various cell cultures, the relationship between nutrient metabolism with epigenetic processes appears universal to at least all mammals [78,85] and is therefore relevant to the study of ruminants.

### 5.2. Transgenerational Effects in Humans and Predictive Adaptive Responses

The case study of the Dutch Hunger Winter provides powerful evidence for the potential for maternal diets to impact the future phenotype of unborn offspring in utero. In 1944, German occupation strictly limited rations available to people living in the western regions of the Netherlands. This severe caloric restriction, followed by prosperity in the post-war period, affected pregnant women of all social classes [86]. This natural experiment revealed that offspring of these pregnant women were disproportionately likely to develop obesity and diabetes compared to both offspring of non-calorie-restricted mothers (other European regions), as well as offspring of calorie-restricted mothers (in Russia) who were themselves also subjected to food shortages later in life [86]. This discovery, along with those of similar natural experiments in the 20th century, led to the development of the previously mentioned “Developmental Origins of Health and Disease” (DOHaD) concept. Within this framework, the “thrifty phenotype” hypothesis was developed, for which strong evidence exists in both humans and rodent models [86]. It was later discovered that, in addition to epigenetic adaptation due to in utero circumstances, offspring can further inherit paternal epigenetic marks passed on through their male germ line [73].

Predictive Adaptive Responses (PARs) are a fundamental concept within the DOHaD paradigm that emphasizes this ability of the developing fetus to modify its physiology and metabolism based on environmental signals during critical periods of development [87,88]. Such responses represent adjustments made by the organism in utero, where environmental cues are interpreted as indicators of the anticipated conditions after birth, and which should optimize survival and reproductive fitness when the prenatal predictions align with the actual postnatal environment [88]. However, if there is a discrepancy or mismatch between the conditions experienced during development and thus anticipated later in life, with those that are actually experienced later, the individual may experience reduced fitness, which manifests, in the case of a “thrifty phenotype”, as an elevated risk of chronic diseases, including type 2 diabetes, obesity, and cardiovascular disorders [89]. In mammals, the maternal and placental environments produce cues that allow the fetus to “read” conditions that might be encountered postnatally. Shifts in those readings promote the selection of alternative developmental paths. When the fidelity of the sensing mechanism is high, the resulting adapted phenotype will possess enhanced fitness. When the prediction is inaccurate, the maladaptation will compromise the organism’s fitness [90]. Although understanding of the mechanisms of this environmentally induced epigenetic inheritance currently remains somewhat ambiguous [55,60], the body of literature supporting transgenerational epigenetic effects in humans, especially as a result of nutrition, is rapidly growing [72,91]. The differentially expressed genes in individuals with a programmed “thrifty phenotype” that underlie their elevated risk for metabolic disease are now understood to include the pancreatic β-cell gene PDX1 [92], metabolism genes PPARα and PPARγ [93], and the growth factor gene IGF2 [94]. While some researchers have studied these effects in the context of dietary phytochemicals [11], such studies in ruminants and other livestock are only beginning to explore the potential implications for animal health and production.

### 5.3. Transgenerational Effects of Nutrition in Ruminants

Within ruminant meat industries, the production of offspring with good health, size, and capacity for growth underlies breeders’ ability to produce marketable livestock. Epigenetic factors, including DNA methylation and long non-coding RNAs, influence many of these traits, from fat deposition [95] to bovine respiratory disease susceptibility [96]. Thus, fetal programming is emerging as an important concept, as it suggests that traits of growth, production, and disease susceptibility are contingent upon the environment their mother is exposed to, especially their diet [58,97]. As with humans, many fetal programming studies have focused on the effect of essential nutrient or calorie deficits. This effect has been long observed in livestock and predates the widespread acknowledgment of epigenetics in the animal sciences. Undernutrition, and in some cases, overnutrition during pregnancy were recognized to cause growth delays in livestock [98] and wild ungulates [99]. More recent studies have begun to connect these phenomena to epigenetic processes [100,101,102] and track their effects in both F1 and F2 generations [101]. Of the macronutrients, protein is especially important for optimal embryonic development. Sufficient protein ensures the establishment of appropriate uteroplacental and fetal vasculature, which allows for the transportation and exchange of respiratory gases, nutrients, and waste products between the maternal and fetal systems. For instance, protein supplementation to cows beginning on day 190 of gestation resulted in a doubling of uterine blood flow when compared with non-supplemented cows [103]. Nutrient transfer to the fetus is critical for establishing adaptive epigenetic marks that will likely persist throughout the offsprings’ lives and may be passed on to their offspring. Indeed, studies have reported increased maturation rates, improved fertility, and earlier calving dates for heifers born to dams supplemented with protein while grazing winter ranges [104,105]. Protein is also critical due to the ability of certain amino acids to function as methyl-group donors. The availability of methyl-group donors during development strongly influences the epigenomes of offspring. Pregnant ewes fed corn, a lower protein feed lacking sufficient levels of certain essential amino acids like lysine, tryptophan, and methionine, gave birth to offspring with a lower methylation of CpG islands when compared to ewes fed alfalfa haylage and ewes fed dried distiller’s grains [106]. Maternal methionine supplementation in dairy cattle resulted in marked changes in offspring gene expression, especially in genes associated with the immune system [83]. A similar study on beef cattle related maternal methionine levels to phenotypic benefits associated with milk production and feed efficiency in mature offspring [80]. This mechanism may explain the observed benefits of milk quality and quantity, as well as animal production from the supplementation of tannins, which bind to rumen proteins, increasing methionine absorption [107,108].

A sufficient intake of nutrients other than protein is also critical. Calves born to cows with access to vitamin and mineral supplements were substantially heavier post-weaning and had improved carcass characteristics and body measurements compared to calves from non-supplemented cows [109]. Interestingly, these differences were not evident in newborn calves, only manifesting later in life (post-weaning), suggesting a delayed DOHaD effect.

## 6. Existing Evidence: Epigenetic Effects of Phytochemicals

### 6.1. The Blurred Primary–Secondary Metabolite Dichotomy

Until recently, secondary metabolites were underexamined relative to primary metabolites, especially in the context of animal health and nutrition. Secondary metabolites are often regarded as being decidedly distinct from primary metabolites, yet this dichotomy is not always well-defined. Phytochemicals were originally termed “secondary compounds” as they were regarded as non-essential for plant growth and survival [15]. However, these compounds maintain plant structure, protect against UV radiation, and minimize water loss, which were likely critical for the emergence and success of vascular plants on Earth [110]. Similarly, from a nutritional standpoint, macronutrients include carbohydrates, fatty acids, amino acids, and nucleic acids, while micronutrients include vitamins and minerals; however, foods and forages likely contain millions of additional compounds that do not fit these categories, far outnumbering primary compounds in diversity [111]. However, this dichotomy is not so clear-cut, considering that phenolic compounds are synthesized from amino acids, such as phenylalanine, with only the removal of an NH3 group distinguishing primary from secondary metabolites [110], and that flavonoids were formerly considered vitamins until being discovered as non-essential in humans [112]. Likewise, carotenoids are non-essential compounds but can serve as direct precursors to vitamin A. Finally, the perception that secondary compounds occur in characteristically minor concentrations relative to primary compounds is not universally true. Phenolic compounds, especially lignin, make up approximately 40% of organic matter in the biosphere [110].

### 6.2. Phytochemicals as Epigenetic Actors

While macronutrients and essential micronutrients govern the major energy metabolism pathways, such as those described above, the contribution of secondary metabolites to epigenetic processes is less well understood and has remained comparatively undervalued. Research has largely focused on common energy-balance metabolites, such as ATP, NAD^+^, Acetyl-CoA, etc.; however, human metabolomes have been estimated to contain anywhere from 3000 to 100,000 compounds [113]. This diversity is not only created by endogenous metabolites from the metabolism of essential nutrients but stems mainly from xenobiotics originating in the consumption (and downstream metabolism) of countless secondary metabolites, especially phytochemicals. Food sources for both humans and ruminants have been historically rich in these compounds [114]. Because secondary metabolites are thought to outnumber primary nutrients by several orders of magnitude, and most of these compounds are unstudied, they are sometimes referred to as the “dark matter of nutrition”. For instance, the metabolite profiles of grass-fed animal-based and plant-based meats differed by 90% despite similar primary nutrient profiles, as described in nutrition fact panels [115]. Livestock grazing diverse pastures accumulate substantially more plant-derived secondary metabolites, leading to marked metabolomic changes when compared to those grazing monocultures or those consuming feed rations [9]. These compounds, especially polyphenols, therefore have significant implications for epigenome regulation [10,11]. However, unlike primary metabolites, phytochemicals are considered non-essential, and their effects on the consumer are highly dose-dependent, making their effects on epigenetic processes more challenging to study.

The phytochemical genistein, a flavonoid common in legumes, has garnered attention for its epigenetic effects. As discussed earlier, genistein’s epigenetic potential has already been demonstrated within the viable yellow agouti mouse model, where fetal exposure to genistein increased methylation, specifically of the IAP promoter, which shifted phenotypes towards the non-agouti and healthy type [76]. Moreover, genistein has been shown to modulate cancer cell growth, sometimes exhibiting prophylactic and therapeutic efficacy against cancer cells both in vitro and within animal models [116,117]. These varying effects have been attributed to genistein’s ability to regulate signal transduction pathways and the enzymatic activity of chromatin modifiers [118]. For instance, genistein has increased H3K4 and H3K9 acetylation and H3K4 di-methylation and decreased H3K9 methylation, all leading to the upregulation of tumor suppressor genes [117]. Genistein can also inhibit DNA methyltransferase (DNMT) enzymes, effectively reversing some of the methylation-caused repression of tumor suppressor genes in cancer cells [117]. Interestingly, genistein has also been shown to have tumor-promoting properties in certain contexts. For instance, genistein sometimes functions as a hormone mimetic, promoting growth in estrogen-dependent breast cancer cells [119].

In addition to genistein, many other phytochemicals, especially polyphenols, can have significant epigenetic effects. Epigallocatechin-3-gallate (EGCG) is a polyphenol abundant within green tea that was found to inhibit DNMT, HDAC, and HAT enzymes [10,11,51]. Thus, the consumption of EGCG has been reported to inhibit proliferation and induce apoptosis of breast cancer cells by modulating the expression of a variety of genes [10,11,51]. Resveratrol, a stilbene common in berries and abundant in the skin of red grapes, is another type of polyphenol that considerable evidence suggests may offer health benefits to humans. Resveratrol can inhibit DMNTs and most HDACs but has an activating effect on SIRT1 within specific pathways. It also decreases the methylated DNA-binding protein MeCP2. Thus, resveratrol upregulates cancer-suppressing genes, inhibiting growth within breast and colorectal cancer cells [10,11,51]. Curcumin is a type of non-flavonoid polyphenol with a wide array of epigenetic effects, including the inhibition of DNMT, HAT, and HDAC enzymes [53,120]. Curcumin was shown to both demethylate the promoter of tumor suppressor genes and inactivate the activity of protooncogenes and pro-metastatic genes, leading to reductions in cancer cells of multiple types [10]. In addition to these more well-documented effects, many other phytochemicals may have—to date—underexplored epigenetic effects. Kaempferol, phloretin, and morin found in apples, apigenin and luteolin found in celery, hesperidin and quercetin found in citrus, caffeic acid and chlorogenic acid found in coffee, sulforaphane found in cruciferous vegetables, allyl mercaptan and diallyl disulfide found in garlic, anthocyanin, piceatannol, and procyanidin found in grapes, and biochanin A, daidzein, and equol found in soy all alter the activity of chromatin modifiers and, ultimately, gene expression patterns, likely providing health benefits in humans when consumed in appropriate doses [10,11]. While we explore epigenetic effects of phytochemicals in ruminants in Section 6.4, the evidence base is currently paltry relative to the human evidence presented above. This discrepancy emphasizes the potential for future research on the ruminant side.

### 6.3. Phytochemicals as Epigenetic Regulators of Immune Response

Research on the epigenetic mechanisms that underlie the medicinal effects of phytochemicals has largely been conducted in the context of human health. While little research has examined the epigenetic effects of phytochemicals in livestock diets, it is reasonable to assume that similar processes to those already identified in humans also occur within other animals. Such processes include the epigenetic regulation of immune response, inflammation, tumorigenesis, and metabolic health, as well as growth rates (Figure 2).

A few recent studies aimed at relating animals’ diets with their epigenomes suggest that these effects may be especially pronounced in genes associated with the immune system. Immune response is a dynamic process that relies largely on epigenetic control in both humans and ruminants [121,122]. Immune cells regularly regenerate, adapt, and store information. For example, an average of 1 × 10^11^ immune cells are renewed per day in the human body [123]. All these cells originate from a common hematopoietic stem cell, which can differentiate into 11 lineages and approximately 100 distinct cell types. DNA methylation, chromatin modifiers, and master transcription regulators are all critical for facilitating the process of hematopoiesis and ensuring it occurs within a relatively stable equilibrium. Epigenetic processes are also important for regulating the inflammatory response and encoding both short- and long-term “cellular memories” of pathogen exposure [121,122].

Dietary phytochemicals may affect immune response via epigenetic mechanisms. Tannins, one of the most common phytochemicals, have been associated with a myriad of health benefits, including improved immune response in livestock [108,124], though the mechanism has not been positively identified. In broiler chickens, tannin supplementation prevented pathogen-associated molecular pattern (PAMP)-induced liver damage by attenuating both innate and adaptive immune responses [125]. Tannin-supplemented broilers had significant decreases in pro-inflammatory cytokines as well as significant increases in immunoglobulins compared to control broilers. Furthermore, researchers demonstrated substantial downregulation of the responsible genes associated with immunological and inflammatory responses known to cause PAMP-related liver damage. A similar study conducted on cattle, sheep, and goat blood revealed that in vitro tannin exposure also altered blood protein concentrations and the transcription of genes associated with both innate and adaptive immunity [126]. Additionally, tannins, when consumed by ruminants, can bind proteins and increase the availability of rumen escape proteins, thereby increasing the availability of methionine (typically the first limiting amino acid from a microbial protein). Methionine is a methyl-group donor that stimulates the production of SAM, a critical cofactor for methylation.

### 6.4. Epigenetic Effects of Phytochemicals in Ruminants

Much of the evidence that phytochemicals can have epigenetic effects in ruminants comes from in vitro experiments. Ruminant blood exposed to tannins in vitro also showed an increased expression of immune-response genes [126]. Recently, the triterpenoid beta-sitosterol was demonstrated to promote bovine preadipocytes differentiation by reducing the expression of the gene *MGP*, thereby activating the PPAR signaling pathway [127]. This has potential implications for aspects of ruminant production, such as efficient fat accrual and deposition. Another recent experiment looked at the effects of a phytochemical on gene expression within the rumen microbial community, rather than in the animal itself. Zhao et al. (2025) found that supplementation of the polyphenol naringin reduced the expression of genes associated with antibiotic resistance in the rumen fluid of Holstein cows [128]. This effect is promising given rising concerns of antibiotic-resistant bacteria due to ruminant production industries.

A few in vivo experiments also provide further evidence within ruminants. Intravenous infusions of the diterpenoid carnosic acid improved oxidative stress biomarkers [129], likely via activation of the PI3K/AKT/Nrf2 signaling pathway [130]. Gabr et al. (2025) recently reported that supplementing goats with a blend of herbal extracts, including oregano, anise, thyme, eucalyptus, and rosemary, combined with *Lactobacilli* improved feed intake, nutrient digestibility, ruminal concentrations of acetic and propionic acids, blood hematological parameters, antioxidant status, and immune response, with a concomitant linear upregulation of the expression of genes associated with growth, immunity, and antioxidant status [131]. A natural bioactive feed additive (the specific phytochemical makeup is proprietary and thereby not reported) attenuated inflammation in healthy Angus heifers via changes in the expression of genes associated with cytokine signaling [132].

### 6.5. Transgenerational Effects of Phytochemicals in Ruminants

Animals might become exposed to primary nutrients or secondary metabolites via three avenues. First, an animal might consume them directly through their feed or forage. Second, a juvenile animal might acquire them through their mother’s milk. Finally, animals might become exposed to them in utero, which could contribute to fetal programming. Any of these modes involve the movement of metabolites from the plant (in the case of herbivores) into the animal, and potentially also movement into the offspring, either through the milk or across the blood–placenta barrier (see Figure 1). Evidence is beginning to demonstrate that these cross-trophic and intergenerational pathways of secondary metabolite movement are not only possible but commonplace.

A collection of recent studies has revealed that potentially health-promoting phytochemicals accumulate in the tissues of grazing animals. A meta-analysis investigating a variety of phytochemicals, including terpenoids, phenols, carotenoids, and tocopherols, described a marked and consistent increase in phytochemical concentrations in grass-fed animals compared with grain-fed animals [9]. This occurred across multiple meat and milk types and increased not only from feedlot to pasture, but from grass monoculture to diverse plant communities [9]. A recent experiment demonstrated differences in 671 out of 1570 profiled compounds between the muscle tissue of pasture-finished and pen-finished bison [133]. Among these different compounds were indicators of improved metabolic health and reduced oxidative stress in the animal, as well as a myriad of potentially health-promoting compounds for consumers of the meat [133]. The presence of phytochemicals in milk has even been attributed to preference for cheeses produced from cows reared on native pasture [134]. Phytochemicals were also found in human breast milk and are purported to be an important source of antioxidants for infants [135,136]. Investigations on the transmission of phytochemicals or other compounds from mother to offspring have largely focused on harmful chemicals, though a few studies have attempted to measure the concentrations of beneficial bioactive compounds that manifest in offspring. Human trials have demonstrated that concentrations of various carotenoids in a mother’s plasma are correlated with concentrations in the mother’s milk as well as infant plasma, suggesting that health-promoting phytochemicals can indeed manifest in offspring through both movement across the blood–placenta barrier and through milk consumption [137,138]. Ewes fed a ration flavored with oregano essential oil during pregnancy produced lambs that were weaned later with a pronounced preference for oregano-flavored feed [139]. This result suggests the movement of phytochemicals, in this example in the form of oregano essential oils, across the blood–placenta barrier, though it is still unclear what role epigenetic processes play in the formation of these distinct dietary preferences. For phytochemicals to pass from mother to offspring, the timing of incorporation in the mother’s diet may be critical. Studies demonstrate that lipid-containing volatile compounds declined precipitously in sheep and beef cattle in relation to the amount of time spent on concentrate feeds [140,141].

While studies outlined in the previous section clearly demonstrate that phytochemicals can have epigenetic effects in ruminants, and evidence outlined in this section suggests some of those effects can become heritable, direct evidence of phytochemicals causing epigenetically mediated transgenerational effects is currently lacking. However, a few studies have examined this topic indirectly. Lan et al. (2013) reported that sheep fed alfalfa hay, known to contain phytochemicals including saponins, during pregnancy reared fetuses with significant differences in gene expression and greater rates of DNA methylation when compared to diets of corn or dried distiller’s grains [106]. Ismail et al. (2025) recently demonstrated that the oral treatment of rams with moringa oil improved markers of semen quality, including sperm motility, viability, and membrane integrity [142]. Furthermore, moringa oil treatments impacted the expression of antioxidant-related genes within ram semen, including the expression of *SOD1* and *CASP3* genes [142]. While this study did not directly evaluate epigenetic marks within germ cells, it demonstrated that dietary phytochemicals can alter gene expression within reproductive cells of ruminants. In vitro exposure of cattle embryos to the phytochemical Coagulansin A reduced embryo DNA damage and decreased the expression of *NF-κB*, attenuating oxidative stress and inflammation, and increased the expression of the heat shock protein gene *HSP70* [143]. This study supports the fetal programming model by which phytochemicals crossing the blood–placenta barrier can induce epigenetic changes in the developing offspring. Still, direct evidence for phytochemically induced epigenetic effects in ruminants inherited transgenerationally is very limited. More research is required to understand the true potential for these effects before strong conclusions can be made.

## 7. Future Directions: Diverse Chemoscapes and Epigenetics

It is often difficult to causally link the observed phenotypic effects of processes such as fetal programming to the specific epigenetic marks that underlie them [71]. This is because epigenetic marks, like DNA methylation, are tissue-specific. They are also temporally dynamic in response to variables such as energy status, time of day, and individual age. An important tool for disentangling these variables is an epigenome-wide association study (EWAS). These studies compare the entire epigenomes of many individuals with those individuals’ phenotypic traits, meaning samples are often required from multiple tissue types and many individuals, making such studies costly and technically challenging. However, human EWASs have been instrumental in identifying epigenetic loci associated with complex traits and diseases [144]. These studies are already being used in ruminant research for applications including the profiling of immune epigenomes in cattle [145], and untold potential exists for using EWASs to identify traits that are sensitive to phytochemical interventions.

A recent and related breakthrough is the discovery of correlated regions of systemic interindividual epigenetic variation (CoRSIVs) in humans [146]. CoRSIVs are genomic regions that reliably differ between individuals yet are reliably consistent between tissues within the same individual [146]. Recently, Chang et al. (2024) described CoRSIVs in cattle as well. CoRSIVs are also associated with regions of genetic variation, suggesting that genetic and epigenetic variations cooperate to determine patterns of interindividual variations in phenotypes. Moreover, DNA methylation patterns within CoRSIVs are considered particularly sensitive to the periconceptional environment, especially nutrition [147]. While the initial analysis by Chang et al. (2024) detected only 217 CoRSIVs in cattle, human research suggests the actual number might be closer to 10,000 [146]. These findings may accelerate the pace of future research because they reveal that body-wide individual epigenetic patterns, once discovered, can be sequenced with only blood samples. Additionally, experiments can be conducted to determine what environmental factors, such as phytochemical exposure, are most influential for forming these patterns during early life.

Native rangelands and other grazing agroecosystems are often characterized by a diversity of plants that provide a broad array of functional phytochemicals. Thus, a promising avenue for future research would focus not only on the singular effects of isolated phytochemicals on animal epigenomes but also on how animal epigenomes respond to the complete phytochemical landscape (chemoscape). Even within classes of phytochemicals, such as polyphenols, specific compounds have distinct chemical structures, and therefore distinct functions and epigenetic effects [40,148]. However, different phytochemicals are also known to interact, both synergistically and antagonistically, affecting animal intake [149]. As discussed in Section 3, landscapes that are higher in functional phytochemical diversity provide ruminants with a greater selection of potential “tools” with which to optimize their health and nutrition. The research explored in this article suggests that epigenetic mechanisms likely mediate such effects. Thus, phytochemically diverse landscapes have implications for not only the animals foraging on them, but also their progeny. Future research promises to uncover more details on the transgenerational effects of dietary phytochemical diversity in livestock.

Multi-omics analyses offer novel methods for examining the relationships between diverse chemoscapes, diet selection processes, and animal metabolomes and epigenomes. Similar analyses are already identifying epigenetically controlled promoters and enhancers associated with key agronomic traits in sheep, such as fat deposition and immunity [150]. As discussed earlier, phytochemically rich and diverse landscapes allow herbivores to optimize their health and maintain homeostasis, encouraging, in turn, the development of preferences for phytochemically diverse diets [149]. Future research might employ multi-omics analyses to investigate this process. The degree to which epigenetic mechanisms underlie both the dietary preferences for and the health benefits from phytochemically diverse diets is not yet fully understood. Indeed, these two processes may interact in a dynamic feedback loop. The combination of epigenomic and metabolomic assays promises to begin drawing a connection from chemoscapes available to animals, to their metabolome, and thereby the epigenome of animals, which may, in turn, influence their dietary preferences, further determining their metabolome and epigenome. Such studies may also offer insight into the degree to which these outcomes are influenced by innate genetic factors, such as breed and species, or epigenetic factors, potentially including in utero influences and later-life learning. However, technical and financial limitations of such approaches will likely remain in the coming years. Because phytochemicals are important but non-essential nutrients that have also been shown to have epigenetic effects, studying the origins and determinants of dietary preference for these compounds may reveal novel insights into the nature–nurture dichotomy. Additionally, because rumen microbial dynamics are so critical in ruminant health and production, future research should be sure to consider how phytochemicals that alter these dynamics can cause epigenetic changes. Ultimately, multiple avenues to research related to the nexus of epigenetics, herbivore nutrition, and plant secondary chemistry are likely to produce meaningful insights in the coming years.

## 8. Conclusions

The production of ruminant livestock is an essential element of the global food supply. However, ruminant production systems face widespread challenges, including animal health and welfare, human nutrition, and environmental sustainability, all while maintaining high levels of animal productivity. Emerging research suggests that the use of phytochemicals may be important in addressing these challenges, despite the general disregard for the role of these compounds as beneficial nutrients in the past. Existing evidence attributes various benefits in the realms of animal health, productivity, and sustainability to appropriate doses of phytochemicals. New evidence suggests epigenetic mechanisms may underlie many of these benefits. If so, it is also possible that some of the benefits observed in ruminants consuming phytochemically rich and diverse diets persist transgenerationally. This could theoretically occur due to early life exposure to phytochemicals that cross the blood–placenta barrier in utero or are transmitted later through milk production. Further, it might occur due to retained or accrued epigenetic marks within parental germ lines. This phenomenon, known as “fetal programming”, has been documented in ruminants, especially in response to maternal nutrition. While many phytochemicals are known to produce epigenetic and phenotypic changes in individuals, few experiments have yet to examine the transgenerational epigenetic effects of phytochemicals, especially in ruminants. This gap emphasizes the need for more research, as the implications would be substantial. Because rearing offspring is the primary function of breeding livestock, any interventions at the breeding and rearing stage that offer persistent benefits in the form of offspring health, productivity, or meat quality would be significant. While the relationship between phytochemicals and offspring traits within ruminant livestock has not been sufficiently investigated, emerging epigenetic research suggests that this is a promising avenue of future research.

## Figures and Tables

**Figure 1 animals-15-01787-f001:**
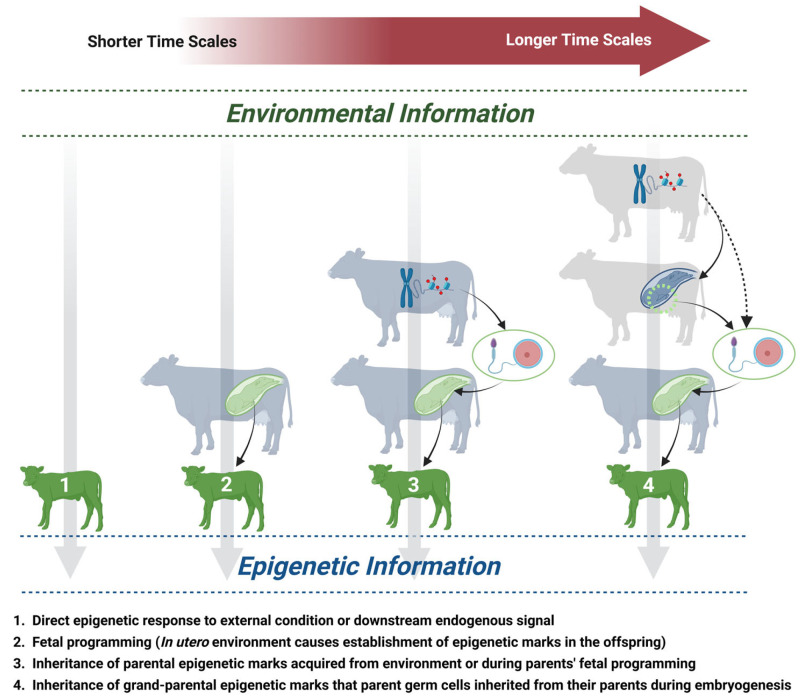
This figure depicts the four principal modes by which epigenetic information can become inherited transgenerationally. These modes are organized from left to right by conveyance of the most recent environmental information to conveyance of the most historically distant information. (1) An epigenetic change might arise in response to a direct environmental condition or a downstream endogenous signal. (2) Fetal programming is a special case of such direct exposure scenarios and occurs commonly during the hypersensitive state of early embryonic development, during which many long-term methylation patterns are being established. (3) An individual can inherit parental epigenetic marks (if those marks persisted through the pre-implantation demethylation phase) that parental germ cells acquired either from their direct environment or during their own fetal programming phase. (4) An individual can inherit grand-parental epigenetic marks (if those marks persist through the pre-implantation demethylation phase, the PGC demethylation phase, and any demethylation that may occur later in life) that parental germ cells inherited from their parents during embryogenesis. Notably, not all changes in epigenetic marks result from environmental information, as some changes that accrue stochastically with age can also be inherited via these four modes. Distinct generations are depicted with the three colors (gray, blue, and green). The potential mechanisms by which environmental signals (e.g., PSCs) are initially transduced into epigenetic marks, as well as key regulatory molecules and gene targets, are depicted in Figure 2. Created in BioRender. Schreiber, S. https://BioRender.com/9hpfx5c accessed on 16 June 2025.

**Figure 2 animals-15-01787-f002:**
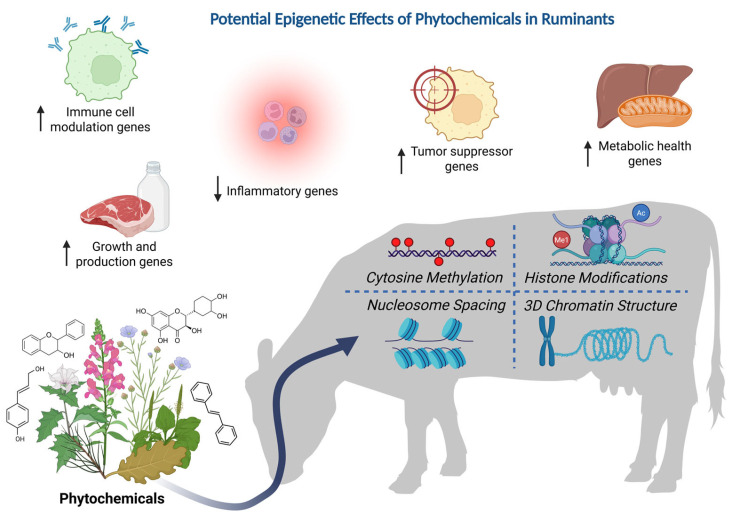
This figure depicts a conceptual framework for the potential epigenetic effects of phytochemicals on ruminant health or production traits. Phytochemicals, when ingested in appropriate amounts, can alter the ruminant epigenome. This manifests through the four layers of epigenetic information depicted within the cow silhouette. This process can have beneficial effects on health and production traits, four primary examples of which are depicted at the top of the figure (based on direct and indirect evidence from human, rodent, and livestock studies). Created in BioRender. Schreiber, S. https://BioRender.com/aaqpcea accessed on 16 June 2025.

## Data Availability

Not applicable.

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
