# Peer review of "Potential Epigenetic Impacts of Phytochemicals on Ruminant Health and Production: Connecting Lines of Evidence"

_animals, 2025, doi:10.3390/ani15121787_

Round 1
Reviewer 1 Report
Comments and Suggestions for Authors
This manuscript explores the frontier topic of how plant secondary compounds (PSCs) may affect ruminant health and transgenerational outcomes through epigenetic mechanisms. The topic is novel and holds significant theoretical and practical value. However, several sections require strengthened evidence, clarification of key arguments, and improvements in terminology consistency and figure clarity. The following comments are provided for the authors’ reference:
-
Title Emphasizes “Ruminant Health and Production” but Relies Heavily on Rodent and Human Data
The manuscript makes extensive use of data from mice and humans (e.g., in Sections 5 and 6), while direct evidence from ruminants remains limited. The authors should consider whether the title needs to be revised to more accurately reflect the content of the review. -
Inconsistent Terminology for PSCs
Terms such as PSC, phytochemicals, plant-derived secondary compounds, and plant secondary compounds are used interchangeably throughout the manuscript. It is recommended that the terminology be standardized for clarity and coherence. -
Inconsistent Definition of Review Type
The review is described as a narrative review, yet the authors mention a “systematic literature search,” which causes ambiguity regarding the review methodology. The review type should be clearly defined and consistently presented. -
Excessive Description of CpG Islands and Histone Modifications
Definitions related to CpG islands and histone modifications are overly detailed and could be condensed to focus on their relevance to the manuscript's core themes. -
Limited Relevance of the Viable Yellow Agouti Mouse Model to Ruminants
The agouti mouse model has contributed significantly to epigenetics, but its direct applicability to ruminant species is questionable. The authors should clearly discuss the extrapolative limitations of using this model in the context of livestock research. -
Section 5: Overreliance on Human and Rodent Studies
The section titled “Existing Evidence: Epigenetic Effects of Nutrition” relies heavily on non-ruminant data, with minimal discussion of studies conducted in cattle, sheep, or goats. This weakens the manuscript’s focus and relevance to ruminant production. -
Lack of Discussion on the Non-Essential Nature of PSCs and Their Dose-Dependent Epigenetic Effects
PSCs are non-essential compounds, and the manuscript does not sufficiently address how this may influence their epigenetic regulatory mechanisms—particularly their potential for dose-dependent or hormetic effects. -
Section 4.2: Overly Detailed Demethylation/Remethylation Pathways
The section on DNA demethylation and remethylation mechanisms is too technical. The authors are encouraged to refocus the discussion on the relevance of these processes to fetal programming in ruminants. -
Section 5.2: Missing Key Gene Examples Related to DOHaD
The discussion on the Developmental Origins of Health and Disease (DOHaD) lacks mention of specific genes that are regulated epigenetically in response to early-life nutrition. -
Section 6.5: Very Limited Direct Evidence for Transgenerational Effects in Ruminants
The potential for PSC-induced transgenerational epigenetic effects is emphasized, but only two studies (Lan et al. 2013 and Ismail et al. 2025) are cited as direct evidence in ruminants. The evidence base is insufficient to support strong conclusions. -
Lack of Integration of Multi-Omics Approaches
The manuscript could benefit from discussion of how multi-omics strategies (e.g., epigenomics combined with metabolomics) can be used to study the “chemoscape” effect of PSCs. Technical and financial limitations of such approaches should also be acknowledged. -
Figures Lack Mechanistic Detail
-
Figure 1 does not clearly illustrate how environmental signals (e.g., PSCs) are transduced into transgenerational epigenetic marks. Key regulatory molecules and gene targets are missing.
-
Figure 2 is too simplified and does not explain how PSCs engage the four epigenetic mechanisms (DNA methylation, histone modification, chromatin remodeling, and non-coding RNAs) to regulate gene expression.
Author Response
Comment 1: Title Emphasizes “Ruminant Health and Production” but Relies Heavily on Rodent and Human Data
The manuscript makes extensive use of data from mice and humans (e.g., in Sections 5 and 6), while direct evidence from ruminants remains limited. The authors should consider whether the title needs to be revised to more accurately reflect the content of the review.
Response 1: Thank you. We agree direct evidence within ruminants remains limited. However, we still believe that the focus of the article is ruminants, and that other lines of evidence are explored to the extent they have relevant implications for ruminants. To reflect this, we have changed the title to “Potential Epigenetic Impacts of Phytochemicals on Ruminant Health and Production: Connecting Lines of Evidence”. Additionally, we have tried to make this clear in our abstract with phrasing such as “Evidence in humans and rodent models, as well as emerging ruminant data have shown phytochemicals can modulate gene expression...”
Comment 2: Inconsistent Terminology for PSCs
Terms such as PSC, phytochemicals, plant-derived secondary compounds, and plant secondary compounds are used interchangeably throughout the manuscript. It is recommended that the terminology be standardized for clarity and coherence.
Response 2: Thank you for pointing this out. We have attempted to standardize the terms by widespread use of “phytochemicals” as per the title, only mentioning PSC initially in the brief paragraph on origins of plant secondary chemistry which we conclude with this section: “In the context of nutrition and potential health benefits, PSCs are often referred to as phytochemicals, which we will use in the remainder of this review. We will also occasionally use the term ‘secondary metabolites’ to refer collectively to phytochemicals and their downstream metabolites within the body of consumers.”
Comment 3: Inconsistent Definition of Review Type
The review is described as a narrative review, yet the authors mention a “systematic literature search,” which causes ambiguity regarding the review methodology. The review type should be clearly defined and consistently presented.
Response 3: Thank you. Multiple other reviewers have pointed out this inconsistency as well. This manuscript was originally drafted as a narrative review. However, upon examination of the instructions for authors for this journal we discovered the following stipulation:
“Narrative reviews should include an Abstract, Introduction, Methods, Relevant sections (body of the narrative review), a Conclusion and Future Directions section, and a References section. The journal requires all review authors to include a Method section, to provide sufficient detail for other scientists to repeat the review. It should specify the search tools used, terms entered, number of hits for each term, and selection process followed.”
This prompted us to include systemic-review-like details in our methods, though we now recognize our phrasing and framing is confusing given the intended style as a narrative review. Please see our updated methods section, where we aim to briefly describe our search protocol (as required per the journal’s instructions) while still being clear about our intent as a narrative review.
Comment 4: Excessive Description of CpG Islands and Histone Modifications
Definitions related to CpG islands and histone modifications are overly detailed and could be condensed to focus on their relevance to the manuscript's core themes.
Response 4: Histone modifications and CpG islands are primarily described in the Foundational Concepts section. We have removed the sentences on CpG islands, but there was only one sentence on histone modifications, and we feel that it is important that researchers without a molecular genetics background understand this concept as many epigenetic effects of PSCs occur by way of HDACs HATs, etc.
Comment 5: Limited Relevance of the Viable Yellow Agouti Mouse Model to Ruminants
The agouti mouse model has contributed significantly to epigenetics, but its direct applicability to ruminant species is questionable. The authors should clearly discuss the extrapolative limitations of using this model in the context of livestock research.
Response 5: We have reduced the word count of this section. Additionally, we have modified the second paragraph to explain the relevance of the agouti mouse model:
“The obvious phenotypic readout made the agouti viable yellow mouse an invaluable and relevant research model to elucidate the basic epigenetic mechanisms, CpG methylation in this case, that connects maternal nutrition to offspring phenotype and ultimately health. While this seminal case study is specific to mice possessing the IAP retrotransposon, the principal that de novo CpG methylation during early development is especially sensitive to environmental conditions, specifically maternal diet, and lead to relatively stable phenotypic changes in the offspring is applicable to all mammals, including ruminants [11,55,66,67]. Furthermore, because supplementation of the phytochemical genistein also induces IAP methylation, this model was amongst the first to demonstrate that phytochemical consumption can cause epigenetic changes, in this case likely via regulation of certain chromatin modifiers [74].”
Comment 6: Section 5: Overreliance on Human and Rodent Studies
The section titled “Existing Evidence: Epigenetic Effects of Nutrition” relies heavily on non-ruminant data, with minimal discussion of studies conducted in cattle, sheep, or goats. This weakens the manuscript’s focus and relevance to ruminant production.
Response 6: Thank you for pointing this out. The aim of section 5.1 is to describe some of the basic pathways active in mammals that relate metabolites with epigenetic outcomes. To help achieve this we cited human studies to support two points – that the metabolome and epigenome are highly correlated, and that SAM serves as a methyl group donor. While both of these points are true likely for all mammals, we agree with your comment and have added ruminant-specific references to support both of those points. We also cite Gut & Verdin 2013 upon several occasions in this section to show how specific intermediate metabolites can modulate gene expression. The Gut & Verdin 2013 paper is in our opinion a seminal review published in Nature that integrates both human data with data from other mammals and cell cultures to produce implications for mammals at large. We believe this is an appropriate reference for elucidating basic mammalian mechanisms which are also at play within ruminants, but that our readers may not be familiar with. The only rodent reference in this section, is to draw a connection back to a previous section on the agouti mouse model, which we believe is a valuable connection for readers to make.
The goal of section 5.2 is to provide a brief academic history of the concepts of transgenerational effects and Predictive Adaptive Responses, which were first described in humans. This section includes well known seminal human studies, which we believe may be helpful for the reader to integrate into their conceptual framework when thinking about how these same effects might occur in ruminants. Because section 5.2 is admittedly human-focused, we have included section 5.3 which exclusively explores how those same concepts bear out in the ruminant literature.
Comment 7: Lack of Discussion on the Non-Essential Nature of PSCs and Their Dose-Dependent Epigenetic Effects
PSCs are non-essential compounds, and the manuscript does not sufficiently address how this may influence their epigenetic regulatory mechanisms—particularly their potential for dose-dependent or hormetic effects.
Response 7: Good point. We have added the following sentence in section 3: “Because the beneficial or harmful effects of PSCs are dose-dependent [Chung et al 1998; Villalba et al. 2017], the concept of “hormesis” is also relevant for understanding this dynamic [Mattson & Cheng 2006].”
We have also added a line referring to their non-essential nature near the beginning of section 3. However, in section 6.1 (The Blurred Primary-Secondary Metabolite Dichotomy), we also try to challenge this notion of a strict and clear dichotomy. We also added the following line to section 6.2: “However, unlike primary metabolites, phytochemicals are considered non-essential, and their effects on the consumer are highly dose-dependent, their effects on epigenetic processes may be more challenging to study.”
Comment 8: Section 4.2: Overly Detailed Demethylation/Remethylation Pathways
The section on DNA demethylation and remethylation mechanisms is too technical. The authors are encouraged to refocus the discussion on the relevance of these processes to fetal programming in ruminants.
Response 8: We have reduced the overall word count, as well as some of the technical jargon within the first 2 paragraphs of this section. Please refer to updated text to see if you have additional recommendation for points to remove. However, we wanted to keep enough text to thoroughly explain the mechanisms at work in figure 1, as you suggest in comment 12, as we are not sure it is possible or preferable to try to explain all of that graphically within the figure itself. Additionally, as you point out in comment 10, there are yet very few studies documenting these different types of effects in ruminants from PSC, so we wanted to at least fully flesh out the mechanisms for those considering conducting future research to fill that gap, since the transgenerational aspect of epigenetics is perhaps the most important part for ruminant industry, given its relace on breeding and offspring production.
Comment 9: Section 5.2: Missing Key Gene Examples Related to DOHaD
The discussion on the Developmental Origins of Health and Disease (DOHaD) lacks mention of specific genes that are regulated epigenetically in response to early-life nutrition.
Response 9: We have added the following sentence to describe some of the many genes at are thought o underlie this phenomenon: “The differentially expressed genes in individuals with a programmed ‘thrifty phenotype” that underlie their elevated risk for metabolic disease are now understood to include pancreatic β-cell gene PDX1 [Vaag et al. 2012], metabolism genes PPARα and PPARγ [Christoforou & Sferruzzi-Perri 2020], and growth factor gene IGF2 [O’Rourke 2014].”
Comment 10: Section 6.5: Very Limited Direct Evidence for Transgenerational Effects in Ruminants
The potential for PSC-induced transgenerational epigenetic effects is emphasized, but only two studies (Lan et al. 2013 and Ismail et al. 2025) are cited as direct evidence in ruminants. The evidence base is insufficient to support strong conclusions.
Response 10: This is true. We have added the following statement at the end of that section: “Still, direct evidence for phytochemically induced epigenetic effects in ruminants inherited transgenerationally are yet very limited. More research is required to understand the true potential for these effects before strong conclusions can be made.” Please let us know if you find other unwarranted conclusions in this section.
Comment 11: Lack of Integration of Multi-Omics Approaches
The manuscript could benefit from discussion of how multi-omics strategies (e.g., epigenomics combined with metabolomics) can be used to study the “chemoscape” effect of PSCs. Technical and financial limitations of such approaches should also be acknowledged.
Response 11: Thank you, we agree multi-omics study are key. We have cited multiple metabolic studies on several occasions within the manuscript to argue the points that 1) metabolomes and epigenomes are highly correlated 2) metabolome diversity is composed largely of secondary metabolites 3) metabolome composition is sensitive to forage type in bovine 4) health promoting phytochemicals can be tracked from forages to the end meat products of beef cattle.
We have also included the following paragraph in section 7 which we believe argues for integrating metabolomics with epigenetics studies more in the future. We also added a statement about technical and financial limitations in this paragraph:
“Multi-omics analyses offer novel methods for examining the relationships between diverse chemoscapes, diet selection processes, and animal metabolomes and epigenomes. Similar analyses are already identifying epigenetically controlled promoters and enhancers associated with key agronomic traits in sheep, such as fat deposition and immunity [141]. As discussed earlier, phytochemically rich and diverse landscapes allow herbivores to optimize their health and maintain homeostasis, in turn, encouraging the development of preferences for phytochemically diverse diets [140]. Future research might employ multi-omics analyses to investigate this process. The degree to which epigenetic mechanisms underlie both the dietary preferences for and the health benefits from phytochemically diverse diets is not yet fully understood. Indeed, these two processes may interact in a dynamic feedback loop. The combination of epigenomic and metabolomic assays promise to begin drawing a connection from chemoscapes available to animals, to their metabolome, and thereby the epigenome of animals, which may, in turn, influence their dietary preferences, further determining their metabolome and epigenome. Such studies may also offer insight into the degree to which these outcomes are influenced by innate genetic factors, such as breed and species, or epigenetic factors, potentially including in utero influences and later-life learning. However, technical and financial limitations of such approaches will likely remain in the coming years. Because phytochemicals are important but non-essential nutrients that are also shown to have epigenetic effects, studying the origins and determinants of dietary preference for these compounds may reveal novel insights to the nature-nurture dichotomy. Ultimately, multiple avenues to research related to the nexus of epigenetics, herbivore nutrition, and plant secondary chemistry are likely to produce meaningful insights in coming years.”
Comment 12: Figures Lack Mechanistic Detail
- Figure 1 does not clearly illustrate how environmental signals (e.g., PSCs) are transduced into transgenerational epigenetic marks. Key regulatory molecules and gene targets are missing.
- Figure 2 is too simplified and does not explain how PSCs engage the four epigenetic mechanisms (DNA methylation, histone modification, chromatin remodeling, and non-coding RNAs) to regulate gene expression.
Response 12: The aim of figure 1 is to graphically represent the modes of transgenerational inheritance of epigenetic information, as explained in section 4.2. The potential mechanisms by which PSCs actually encode information into the epigenome, as well as key regulatory molecules and gene targets are explored in figure 2. We believe for ease of readability it is likely best to separate these two concepts out into two distinct figures.
Figure 2 does not explain precisely how PSCs engage the four epigenetic mechanisms, because based on our review of the literature, this is not yet fully understood. Even so, it is likely that the exact mechanism are diverse and dependent on the exact PSC structure and may not be feasible to depict graphically in this figure. Similarly, we have listed five groups of genes which we expect to be common targets of PSC epigenetic effects, based on the literature on the beneficial effects of PSC in ruminants, and literature on genes that are commonly subject to epigenetic controls in rodent models, humans, and ruminants.
For these reasons, and because none of the other 3 reviewers took issue with the figures, we have decided to modify the figure captions to better communicate the purpose of the figures (higher level & conceptual), while keeping the graphics themselves the same.
Reviewer 2 Report
Comments and Suggestions for Authors
This manuscript titled “Epigenetic Impacts of Phytochemicals on Ruminant Health and Production” discussed the interactions between the fields of plant secondary chemistry, ruminant nutrition, and molecular (epi)genetics. It aimed to familiarize researchers with the scope and foundational concepts of these emerging interactions. It is an innovative perspective that examines the effects of phytochemicals on ruminant health and production from the viewpoint of epigenetic impacts. I have the following suggestions:
It is recommended to add words such as “review” or “perspective” to the 2–3 line title to clearly distinguish that this is a review article.
Lines 40–44 only briefly describe the approach of this article. It is suggested to supplement these with the specific arguments covered in the article.
Regarding the keywords in lines 45–46, the author should consider whether so many are necessary, as some seem not closely related. Also, the term “livestock” should be changed to “ruminant” because the subject of the title and the following text description is ruminants.
In lines 96–98, only some databases were searched. Could this affect the results? Line 103 limits the search to English articles only, which also limits the applicability of the results. It is suggested that the author expand the databases and languages for a more comprehensive summary.
Consider moving the text below Figure 1 in lines 266–172 to the figure legend. In line 704, PSC can be used directly since its full form has appeared earlier in the text. The author should check the whole text for similar issues.
In line 736, the reference 2 lacks volume or page information. In line 1028, the author presentation, Chang, W. -J, is not standardized.
Overall, this is a review with high innovation. However, the title mentions ruminants, but the text seems to focus only on cattle. Also, the impact of phytochemicals on ruminants should mainly be through rumen microbes, and theoretically, there should be differences in the impact on different types of ruminants. These aspects need the author to rethink and reorganize the text.
Author Response
Comment 1: It is recommended to add words such as “review” or “perspective” to the 2–3 line title to clearly distinguish that this is a review article.
Response 1: Based on the comments from another reviewer, we have changed the title to “Potential Epigenetic Impacts of Phytochemicals on Ruminant Health and Production: Connecting Lines of Evidence”. We agree that the former title was not clearly a review, but we believe that this current tile better suggests that the article is indeed a review article.
Comment 2: Lines 40–44 only briefly describe the approach of this article. It is suggested to supplement these with the specific arguments covered in the article.
Response 2: We agree this is a beneficial shift in tone. While heavy “supplementation” is difficult while respecting the requested word limit we have changed the phasing of the lines you references to explain the overarching arguments we make as such: “We argue that heritable epigenetic changes, including “fetal programming”, are commonplace in ruminants under nutritional interventions. We also argue that these phenomena are significant for an industry that relies upon efficient breeding and growth of offspring. We highlight emerging yet limited evidence and offer direction for future research.” Please refer to the updated abstract as leave additional comments if you think this section would benefit from further rephrasing.
Comment 3: Regarding the keywords in lines 45–46, the author should consider whether so many are necessary, as some seem not closely related. Also, the term “livestock” should be changed to “ruminant” because the subject of the title and the following text description is ruminants.
Response 3: We have removed the keywords “foraging behavior” and “grazing” as these are the least related of the key words we included. We also changed “livestock” to “ruminants” in the first sentence of the introduction.
Comment 4: In lines 96–98, only some databases were searched. Could this affect the results? Line 103 limits the search to English articles only, which also limits the applicability of the results. It is suggested that the author expand the databases and languages for a more comprehensive summary.
Response 4: Response 3: Thank you for pointing this out. Other reviewers have pointed out the inconsistency between listing our article as a narrative review (for which extensive methods section are less conventional), and a systemic literature review. This manuscript was originally drafted as a narrative review. However, upon examination of the instructions for authors for this journal we discovered the following stipulation: “The journal requires all review authors to include a Method section, to provide sufficient detail for other scientists to repeat the review. It should specify the search tools used, terms entered, number of hits for each term, and selection process followed.”
This prompted us to include systemic-review-like details in our methods, though we now recognize our phrasing and framing is confusing given the intended style as a narrative review. Thus, in accordance with the other reviewers’ comments, we have simplified, shortened, and re-worded our methods section to reduce this confusion. Please see our updated methods section, where we aim to describe our search protocol briefly (as required per the journal’s instructions) while still being clear about our intent as simply a narrative review.
Comment 5: Consider moving the text below Figure 1 in lines 266–172 to the figure legend. In line 704, PSC can be used directly since its full form has appeared earlier in the text. The author should check the whole text for similar issues.
Response 5: We had not originally included this text in the figure legend to avoid redundancy with the figure itself, since some it is summarized in the numbered list within the figure, but we agree that it may be helpful to have it within the caption as well. We have now included that text within the figure caption.
Comment 6: In line 736, the reference 2 lacks volume or page information. In line 1028, the author presentation, Chang, W. -J, is not standardized.
Response 6: Thank you for catching these errors. We have now fixed them.
Comment 7: Overall, this is a review with high innovation. However, the title mentions ruminants, but the text seems to focus only on cattle. Also, the impact of phytochemicals on ruminants should mainly be through rumen microbes, and theoretically, there should be differences in the impact on different types of ruminants. These aspects need the author to rethink and reorganize the text.
Response 7: Thanks for this comment. Admittedly, while we reference cattle/cows about 18/15 times, and sheep/ewes/rams about 9/4/3 times, and goats only 3 times, this apparent bias is mostly due to a research bias, as the number of relevant studies are highly limited, and all of our search terms used “ruminant” or “livestock” rather than “cattle/sheep/goats”. This is understanding given that the global cattle industry is larger than global small ruminant industries. We have focused on ruminants in this review primarily because 1) they are herbivores and are naturally exposed to a wide variety of phytochemicals through their foraging behavior 2) they are domestic livestock and hold economic importance and 3) their industries rely heavily on reproduction and production of offspring, meaning epigenetic impacts and fetal programming have major implications. This is why we did not focus much on wild ruminants.
We also do not necessarily agree that the impact of phytochemicals on ruminants should mainly be through rumen microbes. While impacts to rumen microbial dynamics are indeed relevant to the health impacts of some phytochemicals, especially proanthocyanidins, it is not clear that it is the primary mechanism by which phytochemicals in general impact ruminant health. If interested, we would be happy to provide some studies or recent reviews to support this statement. Even so, it is not self-evident that rumen microbes being a primary mechanism would mean that the effects of PSC are different among ruminant species, as rumen microbial populations are better predicted by individual diet than ruminant species or breed. Most of the epigenetic mechanisms we discussed are likely basic to the biology of all mammals, not specific to a single ruminant species or breed. Nonetheless, we have made sure to identify whether the studies we reference were conducted on cattle, sheep, or goats, so that readers and other researchers have the opportunity to identify any patterns that are species-specific that we were not able to identify.
For epigenetic effects specifically, rumen microbe interactions may be marginally less critical than for other mechanisms by which PSC can improve ruminant health, because PSC likely impact epigenomes via first entering the circulation, then in tissue-specific metabolomes where they act as signaling molecules. Still, even so, we acknowledge the importance of the rumen microbiome and have attempted to cite all relevant studies that have implicated microbial interactions in the epigenetic effects of PSC. For example, in section 5.3 we wrote about the epigenetic effect proanthocyanidins might have by protecting methionine from rumen microbial degradation, thereby allowing increased methyl donor absorption. In section 6.4, we also explored Zhao et al. 2025, a recent study which looked at the effects of a phytochemical on gene expression within rumen microbial community, rather than in the animal itself. They found that supplementation of naringin reduced expression of genes associated with antibiotic resistance in the rumen fluid of Holstein cows. In the same section we reported on another recent study (Gabr et al. 2025) which showed that supplementing goats with herbal extracts, improved nutrient digestibility, ruminal concentrations of acetic and propionic acids, likely as the result of changes in rumen microbial dynamics.
We have also included the following statement in section 7 to emphasize that we think rumen microbe dynamics should be an ongoing consideration for future research: “Additionally, because rumen microbial dynamics are so critical in ruminant health and production, future research should be sure to consider how phytochemicals that alter these dynamics can cause epigenetic changes.”
Reviewer 3 Report
Comments and Suggestions for Authors
This article surveys how dietary phytochemicals might regulate ruminant health and performance through epigenetic mechanisms. The topic is timely and still emerging; only a handful of primary studies have examined epigenetic endpoints in livestock, so a synthesis could be valuable. The manuscript succeeds in explaining basic epigenetic concepts for an animal-science readership and in highlighting promising research avenues. Nevertheless, important methodological and structural shortcomings prevent the paper, in its present form, from meeting Animals’ standards for a systematic literature review. Major revision is required.
The attached document outlines the areas for improvement.

Author Response
Comment 1:
Issue - Mismatch between stated review type and applied methods
Details - The Methods state that the paper was “structured as narrative review … not based on an exhaustive or otherwise systematic literature search”. Yet elsewhere the authors call it “systematic” and describe a “targeted systematic literature search”.
Action - Specify the review type consistently. If systematic, follow PRISMASR/PRISMA-ScR guidelines; otherwise, re-cast as a narrative scoping review. In particular, it is argued that the present study is better suited to a systematic review. It is therefore recommended that authors adhere to the PRISMA guidelines. This would enhance the transparency of the review process. At present, the number of studies excluded, as well as the reasons for their exclusion, remain unclear. In general, the methods section requires greater levels of detail.
Response 1:
Thank you. Multiple other reviewers have pointed out this inconsistency between our initial claim of a narrative review (for which extensive methods section are less conventional), and a systemic literature review as well. This manuscript was originally drafted as a narrative review. However, upon examination of the instructions for authors for this journal we discovered the following stipulation:
“Narrative reviews should include an Abstract, Introduction, Methods, Relevant sections (body of the narrative review), a Conclusion and Future Directions section, and a References section. The journal requires all review authors to include a Method section, to provide sufficient detail for other scientists to repeat the review. It should specify the search tools used, terms entered, number of hits for each term, and selection process followed.”
This prompted us to include systemic-review-like details in our methods, though we now recognize our phrasing and framing is confusing given the intended style as a narrative review. Other reviewers that noticed this inconsistency suggested reducing the methods to be more consistent with a conventional narrative review, which was our original intention. Thus, in accordance with the other reviewers’ comments, we have simplified, shortened, and re-worded our methods section to reduce this confusion. Please see our updated methods section, where we aim to briefly describe our search protocol (as required per the journal’s instructions) while still being clear about our intent as a narrative review.
Comment 2:
Issue - Search strategy and reproducibility
Details - Only six databases/alerts and seven keyword phrases were used, all in English, with no Boolean logic or MeSH terms shown. Grey literature was excluded without justification.
Action - Supply full search strings, final search dates, language limits and justify exclusions; detail manual searches and alert services.
Response 2: Thank you. These are highly relevant suggestions for a systemic review, but as we have opted to shift the manuscript more in line with a traditional narrative review (per response #1), we are reducing the methods and not including details associated with systemic review methods. Please leave a future comment if you believe some of these are still relevant, but for now we have heeded other reviewers’ suggestions to keep the methods shorter.
Comment 3:
Issue - Study selection, data extraction and critical appraisal absent
Details - Inclusion criteria are listed, but no information is given on duplicate removal, independent screening, disagreements, or the number of records retained at each step. No quality‑assessment tool (e.g., SYRCLE, ROBINS‑I) is applied.
Action - Provide a PRISMA flowchart; describe screening procedures and inter‑reviewer agreement; assess risk of bias; add study‑summary table.
Response 3: Thank you. These are highly relevant suggestions for a systemic review, but as we have opted to shift the manuscript more in line with a traditional narrative review (per response #1), we are reducing the methods and not including details associated with systemic review methods. Please leave a future comment if you believe some of these are still relevant, but for now we have heeded other reviewers’ suggestions to keep the methods shorter.
Comment 4:
Issue - Evidence base too thin for some assertions
Details - Only a handful of animal trials are described; many statements rely on human or other animal’s data.
Action - Temper conclusions; clearly separate ruminant evidence from extrapolations; flag speculative sections.
Response 4: Thank you for pointing this out. Other reviewers have done the same. In section 5.1 we have added some ruminant-specific references to support the human-specific references. At the conclusion of that section we have written: “While this conclusion is supported by lines of evidence from humans, rodents, ruminants, and cell cultures, the relationship between nutrient metabolism with epigenetic processes appears universal to at least all mammals [REF] and is thereby relevant to the study of ruminants.”
Sections 5.2 (Transgenerational Effects in Humans and Predictive Adaptive Responses) and 5.3 (Transgenerational Effects of Nutrition in Ruminants) should be clearly focused on humans and ruminants, respectively. We have also avoiding making any strong conclusions here. Please leave a future comment if you think this is not clear enough.
Section 6.2 (Phytochemicals as Epigenetic Actors) is also based largely on human studies but we have added the following sentence at the end: “While we explore epigenetic effects of phytochemicals in ruminants in section 6.4, the evidence base is currently paltry relative to the human evidence presented above. This discrepancy emphasizes the potential for future research on the ruminant side.”
In section 6.3 (Phytochemicals as Epigenetic Regulators of Immune Response), we have also added some ruminant-specific references when introducing the relationship between epigenome and immune system and avoided strong conclusions.
Essentially all of our conclusions are within section 8 (Conclusion). We have modified this section to avoid strong inferences and reflect the provisional state of the evidence. Please review the updated conclusion section to confirm this.
Comment 5:
Issue - Over‑interpretation in Conclusions
Details - Claims of transgenerational benefits are unsupported by ruminant data.
Action - Rephrase to reflect provisional evidence; emphasize need for multigenerational studies.
Response 5: We appreciate this comment. In the section covering this topic (6.5. Transgenerational Effects of Phytochemicals in Ruminants) we make sure to provide the following caveats:
“While studies outlined in the previous section clearly demonstrate phytochemicals can have epigenic effects in ruminants, and evidence outlined in this section suggests some of those effects can become heritable, direct evidence of phytochemicals causing epigenetically mediated transgenerational effects is currently lacking. However, a few studies have examined this topic indirectly....”
...and at the end we write: “Still, direct evidence for phytochemically induced epigenetic effects in ruminants inherited transgenerationally are yet very limited. More research is required to understand the true potential for these effects before strong conclusions can be made.”
Essentially all of our conclusions are within section 8 (Conclusion). We have modified this section to avoid strong inferences and reflect the provisional state of the evidence. Please review the updated conclusion section to confirm this.
Comment 6:
Issue - Figures and tables
Details - No PRISMA diagram, evidence tables or risk‑of‑bias summary provided.
Action - Add PRISMA flow diagram, evidence tables, and a graphical summary of key pathways.
Response 6: Thank you. These are highly relevant suggestions for a systemic review, but as we have opted to shift the manuscript more in line with a traditional narrative review (per response #1), we are reducing the methods and not including details associated with systemic review methods. Please leave a future comment if you believe some of these are still relevant, but for now we have heeded other reviewers’ suggestions to keep the methods shorter.
Reviewer 4 Report
Comments and Suggestions for Authors
may be corrected as suggested

Author Response
Comment 1: Abstract looks generalized and may be modified as per the information provided in the text
Response 1: Thank you for this comment. We would also like the abstract to include a greater level of detail. Unfortunately, because of the journal’s requirements that abstracts be no longer than approximately 200 words (which we have already exceeded), we can include only 1 sentence or less per (sub)section, and we are limited in the amount of additional information we can provide. Nonetheless, in accordance with your and comments those by other reviewers, we have attempted to reframe some of these sentences to more precisely reflect what we try to accomplish in this manuscript. Please refer to the updated abstract to confirm if it is satisfactory in this regard.
Comment 2: Methods may be condensed
Response 2: Thank you. Other reviewers have expressed similar wishes. While extensive methods sections are not always conventional for narrative review articles, the journal requests that narrative reviews “include a Method section, to provide sufficient detail for other scientists to repeat the review. It should specify the search tools used, terms entered, number of hits for each term, and selection process followed.” Still, we have now simplified, shortened, and re-worded our methods section. Please see our updated methods section, where we aim to briefly describe our search protocol (as required per the journal’s instructions) while still being clear about our intent as a narrative review.
Comment 3: The article is well written and mostly covered Cattle and studies related to buffaloes, sheep and goats may also be included.
Response 3: We appreciate this comment. We did not intend to exclude studies on non-cattle ruminants, and indeed we did include studies conducted on cattle, sheep, goats, and bison. Admittedly, while we reference cattle/cows about 18/15 times, and sheep/ewes/rams about 9/4/3 times, goats only 3 times, and bison once, this apparent bias is mostly due to a research bias, as the number of relevant studies are highly limited, and all of our search terms used “ruminant” or “livestock” rather than “cattle/sheep/goats”. This is understanding given that the global cattle industry is larger than global small ruminant or bison industries.
Comment 4: Key information may be presented as a table
Response 4: Thank you. Some other reviews have mentioned that a table might be appropriate, given that our previous writing of the methods was suggestive of a traditional systemic review, in which case tables summarizing the review process or results of the sources collected are conventional. However, our original intention was for a narrative style review that provides emerging insights and new avenues for research in a topic that it is just starting to be explored in ruminant animals. We have changed our methods section (as well as some other wordings) to better reflect this intention. We believe the important “take aways” or more conceptual in nature and in regard to the potential of future research, and we feel that the current two figures are likely better than a table at depicting these higher level and conceptual take aways. Please leave a future comment if you feel that a table would still be necessary given the manuscript's shift away from a systemic review format.
Comment 5: Impact on health, production and fitness may be explained
Response 5: Thank you. Explaining the epigenetic mechanism that underlie the impacts PSC have on ruminant health and production is of primary importance in this article. In section 3 (Phytochemicals and Animal Health) we explore the general role phytochemicals play in ruminant health. We only had enough space for an introduction to this topic but indeed entire review articles and book could be written and have been written on this topic. Epigenetic mechanisms, which we explore next, are highly relevant to production, because they imply interventions at the breeding and rearing stage can have persistent benefits at later stages of production. We explore the evidence of these impacts to health and production in more detail in sections 5.3 (Transgenerational Effects of Nutrition in Ruminants), 6.3 (Phytochemicals as Epigenetic Regulators of Immune Response), 6.4 (Epigenetic Effects of Phytochemicals in Ruminants), and 6.5 (Transgenerational Effects of Phytochemicals in Ruminants). Common themes are impacts to fat accrual, metabolism, immunity, meat and milk production rates, antioxidant status, and inflammation.
Impacts to animal fitness from PSC was not something we initially explored, as fitness implies long-term evolutionary/selective advantages or disadvantages associated with certain traits. However, we did explore fitness in the context of the phenomenon of fetal programming:
“Such responses represent adjustments made by the organism in utero, where environmental cues are interpreted as indicators of the anticipated conditions after birth, and which should optimize survival and reproductive fitness when the prenatal predictions align with the actual postnatal environment [83]. However, if there is a discrepancy or mismatch between the conditions experienced during development and thus anticipated later in life, with those that are actually experienced later, the individual may experience reduced fitness, which manifest in the case of a “thrifty phenotype” as an elevated risk of chronic diseases, including type 2 diabetes, obesity, and cardiovascular disorders [84]. In mammals, the maternal and placental environments produce cues which allow the fetus to “read” conditions that might be encountered postnatally. Shifts in those readings promote the selection of alternative developmental paths. When the fidelity of the sensing mechanism is high, the resulting adapted phenotype will possess enhanced fitness. When the prediction is inaccurate, the maladaptation will compromise the organism’s fitness [85].”
In accordance with your comment, we have added a statement in section 3 relating PSC consumption to ruminant fitness: “This phenomenon is predicted by co-evolution theory, stating that herbivores that can not only tolerate but derive benefit from low doses of PSCs would have enhanced fitness [Cornell & Hawkins 2003].”
Comment 6: If possible give a note on Epigenome-wide association studies (EWAS)
Response 6: Great point. We agree that these studies are crucial. We have cited an EWAS study by Peterson et al. 2014 on a few occasions within the manuscript. Additionally, we have added a new paragraph to section 7 on EWAS:
“An important tool for disentangling these variables is an Epigenome Wide Association Study (EWAS). These studies compare the entire epigenomes of many individuals with those individuals’ phenotypic traits, meaning samples are often required from multiple tissue types and many individuals, making such studies costly and technically challenging. However, human EWAS have been instrumental in identifying epigenetic loci associated with complex traits and diseases [Campagna et al. 2021]. These studies are already being used in ruminant research for applications including the profiling of immune epigenomes in cattle [Powell et al. 2023], and untold potential exists for using EWAS to identify traits that are sensitive to phytochemical interventions.”
Round 2
Reviewer 3 Report
Comments and Suggestions for Authors
The manuscript has been corrected appropriately.
Reviewer 4 Report
Comments and Suggestions for Authors
the article has been revised as per the suggestions and may be considered for publication